# Structural and Photophysical Properties of 2,1,3-Benzothiadiazole-Based Phosph(III)azane and Its Complexes

**DOI:** 10.3390/molecules25102428

**Published:** 2020-05-22

**Authors:** Radmir Khisamov, Taisiya Sukhikh, Denis Bashirov, Alexey Ryadun, Sergey Konchenko

**Affiliations:** Nikolaev Institute of Inorganic Chemistry, Siberian Branch of the Russian Academy of Sciences, 630090 Novosibirsk, Russia; khisrm@gmail.com (R.K.); bashirov@niic.nsc.ru (D.B.); ryadunalexey@mail.ru (A.R.); konch@niic.nsc.ru (S.K.)

**Keywords:** 4-amino-2,1,3-benzothiadiazole, phosph(III)azane, cyclophosp(III)azane, fluorescence, coordination compounds, single crystal X-ray diffraction

## Abstract

Here we describe the synthesis of a novel N,N’-bis(2,1,3-benzothiadiazol-4-yl)-1-phenylphosphanediamine (H_2_L) and its zinc (II) and copper (I) coordination compounds [Zn_2_L_2_]·nC_7_H_8_ (**1**·nC_7_H_8_), [Zn_2_(H_2_L)_2_Cl_4_]·nC_7_H_8_ (**2**·nC_7_H_8_), and [Cu(H_2_L)Cl]_n_·nTHF (**3**·THF). According to single crystal X-ray diffraction analysis, H_2_L ligand and its deprotonated species exhibit different coordination modes. An interesting isomerism is observed for the complexes [Zn_2_(H_2_L)_2_Cl_4_] (**2a** and **2b**) that differ by the arrangement of H_2_L. Both complexes possess internal cavities capable of incorporating toluene molecules. Upon toluene release, the geometry of **2b** changes substantially, while that of **2a** changes slightly. Due to the diverse structures, the compounds **1**–**3** reveal different photophysical properties. These results are discussed based on previously reported studies and DFT (density functional theory) calculations.

## 1. Introduction

Photoluminescent organic compounds and transition metal complexes with organic ligands are important components in various scientific applications, including optoelectronics, photocatalysis, and photodynamic therapy. Heteroaromatic units, in particular 2,1,3-benzothiadiazole (btd) derivatives, are often used as building blocks of optical materials [1,2,3]. Generally, btd acts as an acceptor in polymer molecules [4,5]. However, btd-based small molecules are also being actively studied [6,7,8,9,10]. The photophysical behavior of such derivatives can be tuned by two ways; (1) functionalization of btd; (2) coordination to suitable metal ions. From the viewpoint of coordination chemistry, the high binding affinity of ligands towards transition metals is very important. By choosing appropriate donor groups capable to binding with metal, one can construct defined species with particular ligand arrangement. The presence of different types of donor atoms in one ligand opens up the possibility to interplay with various metal ions as well as to perform a wide variety of coordination modes of this ligand [11]. In this context, P,N-containing ligands are highly attractive, as they combine both strong and weak Lewis base centers. A number of btd derivatives bearing N-donor functional groups were recently studied [2,10,11,12,13]. Phosphorus-containing btds are much less abundant; they are mainly limited to derivatives with phosphate groups [14,15,16,17,18].

Compared with btds, the luminescence properties of organophosphorus derivatives are relatively poorly explored. However, the latter have been studied as optical materials in the last decade. Phosphorus (V) compounds have been tested as acceptor blocks and hosts in optical materials [19,20,21], while reports on P(III)-based compounds in the field of photophysical applications [22], e.g., in organic electronics [23] are scarce, due to their air-sensitivity. Meanwhile, phosphorus (III) derivatives are interesting objects in terms of the ability of the P atom to coordinate metals, allowing the design of various coordination compounds. In fact, coordination of a metal protects P atom from its interaction with oxygen, which increases the stability of a complex to oxidize with respect to free P(III) derivative.

Recently, we reported the study of the phosphorylated 4-amino-2,1,3-benzothiadiazole derivative − N,N-bis(diphenylphosphanyl)-2,1,3-benzothiadiazol-4-amine ((Ph_2_P)_2_N-btd) (Scheme 1) and its transition-metal complexes [24]. This ligand contains one btd moiety and tends to coordinate in a bidentate manner via two P atoms. In the present work, we synthesized a novel amino-benzothiadiazole derivative bearing two secondary amino groups and a phosphino group, namely, N,N’-bis(2,1,3-benzothiadiazol-4-yl)-1-phenylphosphanediamine (H_2_L). Contrary to (Ph_2_P)_2_N-btd, H_2_L contains acidic hydrogen atoms and thus can act as anionic ligand. In addition, H_2_L bears two btd moieties that provide a possibility to bridging coordination via two N atoms. The coordination chemistry of H_2_L was demonstrated by means of few examples of its metal complexes, viz. Cu(I) and Zn(II). These central metals are frequently used for luminescent applications as earth-abounded non-precious transition elements. Structural aspects of the compounds prepared were studied using single crystal X-ray diffraction (XRD) analysis and quantum chemical calculations. Photophysical properties of the compounds were also studied.

## 2. Results and Discussion

### 2.1. Synthesis

A novel N,N′-bis(2,1,3-benzothiadiazol-4-yl)-1-phenylphosphanediamine (H_2_L) can be prepared from 4-amino-2,1,3-benzothiadiazole (NH_2_-btd) by its modification at amino group using phosphorylation method. The reaction of the benzothiadiazole derivative and dichlorophenylphosphine in a mole ratio 2:1 in the presence of triethylamine gave the title phosphazane in the 80% isolated yield (Scheme 1). The phosphorus reagent, as well as the reaction product, is water and air sensitive; therefore, the reactions were performed under strong anaerobic conditions. The product as a yellow crystalline solid was purified of unreacted amine, phosphine and other by-products by washing with diethyl ether, as the target phosphazane is poorly soluble in it. One of the by-products, namely N-[1,3-bis(2,1,3-benzothiadiazol-4-yl)-2,4-diphenyl-1,3,2λ^5^,4-diazadiphosphetidin-2-ylidene]-2,1,3-benzothiadiazol-4-amine (**4**) was isolated by cooling of the ether solution as a few crystals analyzed by single crystal XRD (Figure 1). It can be assumed that the formation of **4** is a result of a side condensation reaction [25]. Most probably, phosph(III)azene PhP=Nbtd was formed and condensed with H_2_L in the reaction mixture. To establish the exact formation pathway of **4**, a further research is required.

H_2_L contains two acidic hydrogens and thus can act as one- or dibasic acid in the presence of strong bases. Thus, we studied the coordination chemistry of the phosphazane behaving as both anionic and neutral ligand.

The complex [Zn_2_L_2_] (**1**) was obtained by treatment of H_2_L with ZnCl_2_ and KHMDS (potassium bis(trimethylsilyl)amide) in THF in a mole ratio 1:1:2 (Scheme 2) as dark red powder. The complex was recrystallized from toluene and isolated as its solvate [Zn_2_L_2_]·nC_7_H_8_ (**1**·nC_7_H_8_) with the yield of 40%. In **1**, the ligand L^2–^ is twice deprotonated. We also attempted to synthesize a complex with partially deprotonated HL^−^ ligand by using other reagents ratio, namely H_2_L:KHMDS = 1:1. However, the reaction gave the complex **1**. Upon drying in vacuum, the compound loses almost all lattice toluene; according to the elemental analysis, its content n equals to circa 0.1.

The synthesis of zinc (II) and copper (I) complexes with neutral H_2_L was achieved by the treatment of the phosphazane with ZnCl_2_ and CuCl, respectively (Scheme 2). The high solubility of the zinc complex in THF prevented the obtaining of crystals suitable for XRD analysis. Therefore, in the synthesis of the zinc complex, the solvent was changed to toluene, in which it is slightly soluble. The slightly different habitus of the two types of crystals allowed for the harvesting of the samples for single crystal XRD analysis. According to the data, two isomeric complexes of the formula [Zn_2_(H_2_L)_2_Cl_2_]·nC_7_H_8_ (**2a**·3C_7_H_8_ and **2b**·2.5C_7_H_8_) were formed. Having similar chemical-physical properties, the compounds were not separated from each other. The total yield was 95%. Upon drying, part of the solvate molecules is lost, resulting in a structural transformation of both compounds (Appendix A). The elemental analysis data of the mixture of phases (**2**·nC_7_H_8_) indicate the toluene content n of 1.1.

The complex [Cu(H_2_L)Cl]_n_·nTHF (**3**·THF) is insoluble in THF. To obtain a crystalline product, the reaction was carried out without stirring by adding a solution of H_2_L in THF to a solid CuCl. After a week, the heterogeneous reaction was completed with crystalline complex **3**·THF formation. The isolated yield was 60%. According to XRD analysis, the complex is polymeric, which explains its insolubility in THF. Both XRD and elemental analysis indicate the compound has one THF per formula unit.

### 2.2. Structural Characterization

Figure 2 shows the crystal structure of H_2_L. Both amine nitrogen-centered fragments N^4^ and N^4′^ (for atom numbering, see Scheme 2) have a trigonal planar geometry assuming strong conjugation of the lone pairs of the N atoms with the aromatic system of the btd. The molecule H_2_L approximately belongs to *C_s_* symmetry (Schoenflies notation) with the mirror plane going through P atom. Small differences in the arrangement of btd moieties relative to PPh violate this symmetry. For instance, the P–N–C valent angles equal to 127.3° and 125.7°, while those of P–N–H equal to 124° and 128°. The torsion angles C–P–N–C slightly differ from one another (Table 1, Appendix A). The DFT-optimized molecule has similar geometry with the symmetry closer to *C_s_* (Table 1).

The crystal packing of H_2_L shows N···S intermolecular interactions between btd moieties (Figure 3). The intermolecular distances between adjacent N and S atoms lie in the range 3.04–3.51 Å. These distances are consistent with the literature data for secondary bonding interactions between btd moieties [24,26,27]. In H_2_L, two crystallographically independent types of N···S contacts are observed: the first type links the molecules into a chain via single interactions (Figure 3, on the right), while the second links via double ones (Figure 3, on the left).

XRD molecular structure of **1** is shown in Figure 4. The compound **1** is a binuclear complex in which zinc atoms are linked by two chelate-bridging L^2–^ ligands. The coordination environment of Zn is tetrahedral. In accordance with the strong and weak Lewis acids and bases concept, the ligands coordinate the metals via N^3^ and N^4^ atoms, while P atoms remain free. The complex has *C*_2_ symmetry with two-fold axis going parallel to two btd moieties. N^4^ atoms of deprotonated amine groups have trigonal planar coordination environment of P, C and Zn atoms. One of C–P–N–C torsion angles significantly differ by circa 70° from that of free H_2_L; this implies that the HN-btd unit can rotate freely along the P–N bond adjusting to the metal environment. The N–P–N angle is smaller than in H_2_L, likely due to the attraction interaction of N^4^ with the neighboring Zn atom at a distance of 3.03 Å (Appendix A). Two out of four btd moieties enter a π-π stacking between two ligands in the complex molecule.

The isomeric complexes **2a**·3C_7_H_8_ and **2b**·2.5C_7_H_8_ differ from each other by the ligand arrangement (Figure 5). In **2a**·2.5C_7_H_8_, the btd fragments oriented in different directions (head-to-tail manner), while in **2b**·2.5C_7_H_8_, they oriented in one direction (head-to-head manner). As a result, the first complex molecule has *C_2h_* symmetry, while the second one has *C_2_* symmetry. Zn atoms reveal similar tetrahedral coordination environment of two N and two Cl atoms. In contrast to L^2–^ in **1**·C_7_H_8_, H_2_L acts as a bridging ligand, coordinating only via N^1^ and N^1′^ atoms. Geometry of the ligand in the complexes resembles that in free H_2_L. The complex molecules in both structures have tetragonal prismatic cavities filled by toluene molecules (Figure 6, top). The phenyl planes of the toluene molecules are not arranged parallel to any btd aromatic system. The corresponding planes are perpendicular to P···P (in the case of **2a**·3C_7_H_8_) or Zn···Zn (in the case of **2b**·2.5C_7_H_8_) diagonals within the complex molecules. The particular arrangement of the toluene causes quantitative differences in the geometry of the cavities. The structure of **2b**·2.5C_7_H_8_ reveals quite similar Zn···Zn (of 11.6 Å) and P···P (of 11.4 Å) intramolecular distances, while those in **2a**·3C_7_H_8_ are different, being 12.9 and 10.4 Å respectively (Figure 6, top). In the structure of **2a**·3C_7_H_8_, the solvent molecule is disordered over four positions due to proximity to the two-fold rotation axis and the mirror plane, while in **2b**·2.5C_7_H_8_, the molecule has a single position.

Upon DFT-optimization in the absence of the toluene molecule, both molecules **2a** and **2b** exhibit distorted geometry with elongated Zn···Zn and shortened P···P distances (Figure 6, bottom). The main structural changes occur due to reducing N–Zn–N angles (Table 1). We conclude that the geometry of **2b** changes more strongly with the removal of the toluene than that of **2a**, while the cavities of free **2a** and **2b** become similar. This implies that the particular conformation of complex **2a** with similar Zn···Zn and P···P diagonal distances is stabilized by the inclusion of the toluene molecule, but it is not related to the “head-to-tail” arrangement of the ligands. The total energy of **2a** and **2b** is similar (the difference of 4.10 kJ/mol), which suggests equally probable formation of the “head-to-head” or “head-to-tail” isomers from a solution. Analysis of Hirshfeld surfaces [28] of the crystal structures did not revealed the presence of specific interactions between the complexes and toluene (Appendix A). In principle, other molecules forming specific interactions with the complex could enter the cavities. This is a possible way of stabilizing one or another isomer, which allows one to isolate **2a** or **2b** as a single phase.

The difference in geometry of the molecules leads to different packing in the crystal. The complex molecules in **2a**·3C_7_H_8_ are located one above another and form stacks along *c* direction (Figure 7). In contrast, in **2b** molecules are staggered along *c* direction (Figure 8).

Unlike the abovementioned molecular complexes, complex **3**·THF is a coordination polymer (Figure 9). Cu(I) as a weak Lewis acid has P atom in its environment; further coordination places are occupied by one N and two Cl atoms, forming a tetrahedral coordination polyhedron. Both H_2_L and Cl^−^ ligands act as bridging ones, thus forming a polymer chain. btd moieties not bonded directly to copper atoms enter interchain π-stacking (Figure 10). Notably, the conformation of H_2_L ligand in **3** significantly deviates from that of free H_2_L with reduced C–P–N–C and P–N–C–C(N) angles (by circa 150° and 35° respectively; Table 1). This conformation is obviously implemented to avoid unfavorable Cu···HC repulsive interactions (Appendix A).

The cyclodiphosphazane **4** has an unusual structure; it formally contains both P(V) and P(III) atoms in a four-membered P,N-cycle (Figure 11). In general, such cyclophosphazanes contain either trivalent [29] or pentavalent [30] phosphorus atoms. The P(V)–N bonds (of 1.69 Å) in the cycle are shorter by 0.08 Å compared to the P(III)–N ones. The P(V) atom attaches a pendent N-btd fragment; the corresponding P(V)–N bond is the shortest (of 1.54 Å), indicating the absence of hydrogen at the N atom. The corresponding bonds agree well with the literature data (see, for instance, those in *t*-butylamido-arylimino-derivatives [31]) and with the DFT-optimized molecule. Two btd fragments are located nearly in the same plane with the P,N-cycle. In the crystal packing, an intermolecular π- stacking is observed between these btd fragments.

### 2.3. Photophysical Properties

For all compounds in the solid state (polycrystalline samples for luminescence studies and a mixture with BaSO_4_ for UV-vis), the absorption (Figure 12) and photoluminescence (Figure 13) spectra were measured. Only H_2_L and **2**·nC_7_H_8_ show radiation in the visible range, while **1**·nC_7_H_8_ and **3**·THF do not emit in this range. For the luminescent compounds, lifetimes and quantum yields were also measured. All investigated spectroscopic and photophysical data are summarized in Table 2.

The UV-vis spectra of H_2_L, **1-3** in the solid state have similar peaks at around 305 nm. H_2_L, **2**·nC_7_H_8_ and **3**·THF have a long-wavelength absorption band lying in the range of 380–480 nm. Compared to H_2_L, this band is bathochromically shifted in the case of compounds **2**·nC_7_H_8_, while it is hypsochromically shifted for compound **3**·THF, although the latter features a broad low-intensity shoulder spanning the range of 500–600 nm. This behavior can be explained by different coordination types of H_2_L. Recently we have shown that for compounds with NH_2_-btd and (Ph_2_P)_2_N-btd ligands coordinated via N^1^ atom, the absorption and emission band maxima are bathochromically shifted compared to the corresponding free btds [11,24]. This tendency is observed for a number of Cu(I), Zn(II), Ag(I), Cd(II), Pd(II) and Pt(II) complexes studied; the nature of the metal and conformation of the ligand affects the position of the bands to a lesser extent. Compound **3**·THF somewhat breaks the strong correlation between the coordination via N^1^ and the bathochromic shift (if the shoulder is not considering). Probably, this is due to cooperative effects of significantly distorted conformation of the ligand compared to free H_2_L and the presence of the uncoordinated btd unit, which manifests itself in a hypsochromic shift and/or a decrease in the intensity of the band.

The absorption spectrum of compound **1**·nC_7_H_8_ reveals two absorption bands in the visible region. The band peaked at 600 nm is the most long-wavelength one among the compounds. Possible reasons of this behavior are as follows: (1) contrary to the others, the ligand in **1**·nC_7_H_8_ is deprotonated and thus possess a different electronic structure; (2) the coordination of the ligand via N^3^ atom can result in a bathochromic shift of the band. To date, only two compounds with anionic ligand containing the btd-N^−^ unit and their photophysical studies have been reported, namely, Zn and Sm complexes with 4-(2,1,3-benzothiadiazol-4-ylamino)pent-3-en-2-onate [32]. They fit to the second proposed hypothesis: the Zn complex reveals coordination of the ligand only via O and N^4^ atoms and exhibits position of the absorption and emission bands similar to free neutral btd derivative. In the Sm complex, the ligand further coordinates via N^3^, while the emission band is bathochromically shifted, as in the case of **1**·nC_7_H_8_. TD-DFT (vide infra) calculations reveal the frontier molecular orbitals in complex **1** significantly differ from the orbitals in the H_2_L and **2**.

According the TD-DFT calculation on B3LYP theory level, the absorption band of H_2_L at 305 nm corresponds to a transition between the HOMO and LUMO + 2, which can be described as a charge transfer from btd to PPh fragments (Appendix A). Long-wavelength bands mainly correspond to the HOMO → LUMO (452.1 nm), HOMO → LUMO + 1 (436.8 nm), HOMO – 1 → LUMO (402.3 nm) and HOMO – 1 → LUMO + 1 (398.5 nm) transitions. According to the depicted frontier molecular orbitals (Figure 14), the LUMO and LUMO + 1 orbitals are preferably located on the btd fragments, while the HOMO and HOMO – 1 orbitals have significant localization on the amino groups. Consequently, the low-energy transitions can be described as a charge transfer from the amino group to the btd fragments. The position of the long wavelength absorption and emission bands of H_2_L and (Ph_2_P)_2_N-btd is similar, although the latter is rather characterized by a charge transfer from PPh_2_ to btd [24].

According TD-DFT calculations, the low-energy bands of the coordination compounds arise from ligand-centered transitions. In **1**, three long-wavelength transitions mainly correspond to the promotions HOMO → LUMO (671.4 nm), HOMO → LUMO + 1 (629.3 nm) and HOMO → LUMO + 2 (623.8 nm) (Appendix A). Contrary to free H_2_L, different btd moieties of each of the two ligands contribute to the orbitals (Figure 15, Appendix A): LUMO orbitals are preferably localized at the stacked btd moieties and HOMO—at PPh and pendant btd moieties. This is manifested in a large charge transfer. In **2b**, three first intense transitions correspond to the HOMO → LUMO (532.2 nm), HOMO → LUMO + 1 (506.8 nm) and HOMO – 1 → LUMO (503.7 nm) (Appendix A). They are also characterized by charge transfer; however, the orbitals resemble that of free H_2_L: the LUMO orbitals are mainly localized on the thiadiazole moiety, while the HOMO orbitals—on the amino group and the benzene ring of btd. Thus, as expected, the bathochromic shift of the long-wavelength band in **1** and **2b** occurs due to a decrease of the HOMO-LUMO gap from 3.28 eV for H_2_L to 2.35 eV for **1** and 2.85 eV for **2b**. We assume that **2a** shows similar to **2b** electronic structure and thus shows similar photophysics; for this reason, TD-DFT calculations for **2a** were not performed. The calculations for **3** were also not performed due to its complicated polymeric structure.

The emission band of the compounds **2**·nC_7_H_8_ is significantly bathochromically shifted by 2770 cm^−1^ compared to H_2_L. The excitation spectra are slightly batchochromically shifted compared to the corresponding absorption spectra. The absolute quantum yield of emission (QY) upon transition from H_2_L to the complex decreases from 8% to 3% (Table 2). The photoluminescence lifetimes for H_2_L and **2**·nC_7_H_8_ belong to the nanosecond time scales, indicating that emission occurs by fluorescence mechanism from the singlet excited state (Table 2). Note that the mixture of compounds **2**·nC_7_H_8_ has both the single-peak band and the single-exponential decay kinetics. This means that either these isomers exhibit similar photophysical properties, or one of them has a much lower emission intensity compared to the other.

## 3. Materials and Methods

### 3.1. General Methods

Starting materials, except for solvents, were used as received from suppliers. 4-amino-2,1,3-benzothiadiazole (4-NH_2_-btd) [33] was synthesized as reported previously. The synthesis of novel compounds was carried out in evacuated vessels by using Schlenk techniques at room temperature unless otherwise specified. The solvents were distilled in inert atmosphere over common drying agents. Elemental analysis was performed with a Eurovector EuroEA3000 analyzer (Eurovector SPA, Redavalle, Italy). ^1^H NMR spectra (500.13 MHz) and ^31^P NMR spectra (202.45 MHz) were taken with a Bruker DRX-500 spectrometer (Bruker Corporation, Billerica, MA, USA) in C_6_D_6_ at room temperature; the solvent peak was used as internal reference. The IR spectra were recorded in KBr pellets at room temperature by means of a FT-801 Fourier spectrometer (Simex, Novosibirsk, Russia). The diffuse reflectance UV-vis spectra were obtained with a Shimadzu UV-3101 spectrophotometer (Shimadzu, Kioto, Japan) at room temperature. Samples for the diffuse reflectance measurements were prepared by a thorough grinding of a mixture of the compounds under study (about 0.005 mole fraction) with BaSO_4_, which was used also as a standard. Spectral dependences of the diffuse reflectance were converted into spectra of a Kubelka–Munk function [34]. Emission and excitation spectra were recorded on a Fluorolog 3 spectrometer (Horiba Jobin Yvon, Edison, NJ, USA) equipped with cooled PC177CE-010 photon detection module with a PMT R2658 photomultiplier. Excitation and emission spectra were corrected for source intensity (lamp and grating) and emission spectral response (detector and grating) by standard correction curves. For the measurements, powdered samples were placed between two nonfluorescent quartz plates. Absolute quantum yields were determined using Quanta-phi integrating sphere (Horiba Jobin Yvon, Edison, NJ, USA). Luminescence decay kinetics was recorded by a time-correlated single photon counting (TCSPC) technique using a NanoLED pulsed light source and a NanoLED-C2 controller on a Fluorolog 3 spectrometer (Horiba Jobin Yvon, Edison, NJ, USA).

Hirshfeld promolecular surface mapped over d_norm_ plots of the complexes were built using Crystal Explorer (version 17.5) program [28].

### 3.2. Quantum Chemical Calculations

Quantum chemical calculations were performed using the ORCA v. 4.2.1 computational package [35,36].

The full geometry optimization of H_2_L, **1**, **2a**, **2b** and **4** molecules was carried in the gas phase without symmetry constraint at DFT level with B3LYP functional and Ahlrichs def2-SVP basis set. Atom-pairwise dispersion correction D3 with Becke–Johnson damping were used for geometry optimization [37]. Optimized geometries were submitted to numerical frequency analysis at the same level of theory to check whether the stationary point were “genuine” minima; no imaginary frequencies were found. The resolution of identity chain-of-spheres module, RIJCOSX was used to reduce the computational cost of the calculations [38].

We performed several TD-DFT H_2_L excitation energies calculations with various functionals in the gas phase and def2-TZVPP basis set. B3LYP, B97XD, BH&HLYP and CAM-B3LYP. B97XD, BH&HLYP and CAM-B3LYP gave poor results, underestimating the transitions energy (Appendix A). The B3LYP calculation results were closer to the experimental data. For **1** and **2b**, TD-DFT calculations were performed with B3LYP functional and def2-TZVPP basis set in the gas phase. 

### 3.3. X-ray Structure Determination

Single crystal XRD data for the compounds H_2_L, **1**–**3** were collected with a Bruker Apex DUO diffractometer (Bruker Corporation, Billerica, MA, USA) equipped with a 4K CCD area detector and a graphite-monochromated sealed tube (Mo Kα radiation, λ = 0.71073 Å). The data for **4** were collected with a Bruker D8 Venture diffractometer (Bruker Corporation, Billerica, MA, USA) with a CMOS PHOTON III detector and IµS 3.0 source (Mo Kα radiation) (Appendix A). All measurements were conducted at 150 K, the φ- and ω-scan techniques were employed. Absorption corrections were applied with the use of the SADABS program [39]. The crystal structures were solved using the SHELXT [40] and were refined using SHELXL [41] programs with OLEX2 GUI [42]. Atomic displacement parameters for non-hydrogen atoms were refined anisotropically, with the exception of some disordered molecules. Hydrogen atoms were refined in riding model with the exception of those of the amino group, which were refined freely with the DFIX restraint on the corresponding N–H bonds. Phenyl groups in **2b**·2C_7_H_8_ were disordered over two positions due to the proximity to the 2-fold rotation axis.

CCDC 1996456–1996460 and 1998636 contain the supplementary crystallographic data for this paper. These data can be obtained free of charge from the Cambridge Crystallographic Data Centre via www.ccdc.cam.ac.uk/data_request/cif [43].

The powder XRD analysis of the compounds was performed using a Shimadzu XRD-7000 diffractometer at room temperature (CuK_α_ radiation, Ni filter) (Shimadzu, Kioto, Japan) Powder samples were slightly ground with heptane in an agate mortar and deposited on the polished side of a quartz-glass holder. Powder diffraction patterns were collected in 5–30° 2θ range.

### 3.4. Syntheses

#### 3.4.1. Synthesis of H_2_L

A solution of PhPCl_2_ (0.179 mL, 1.32 mmol) in 10 mL of toluene was slowly added to a cooled to 0 °C solution of 4-amino-2,1,3-benzothiadiazole (0.400 g, 2.65 mmol) and Et_3_N (0.6 mL, 4.3 mmol) in 10 mL of toluene. A precipitate formed gradually in the solution. The resulting suspension was stirred overnight, followed by filtration to remove triethylamine hydrochloride. Slow evaporation of the solvent gave yellow plate crystals. The crystalline product was washed with ether (3 × 2 mL) and dried under vacuum. Yield 0.420 g (78%). Calc. for C_18_H_13_N_6_PS_2_ (%): C 52.9; H 3.3; N 20,0; found: C 52.9; H 3.2; N 20.6. ^1^H NMR (C_6_D_6_, δ, ppm): 6.37 (s, 2H); 7.08 (m, 2H); 7,15–7,21 (m, 3H, Ph-H + solvent); 7,23 (s, 1H); 7,26 (dd, 2H); 7.38 (d, 2H); 7,66–7,69 (m, 2H). ^31^P{H} NMR (C_6_D_6_, δ, ppm): 43,5 (s). IR (cm^−1^): 3337 (m), 3049 (w), 2922 (w), 2851 (w), 1972 (w), 1911 (w), 1829 (w), 1718 (w), 1604 (m), 1547 (s), 1484 (s), 1437 (s), 1373 (s), 1274 (s), 1189 (w), 1161 (w), 1082 (s), 1038 (m), 999 (w), 905 (s), 849 (s), 808 (s), 740 (s), 703 (m), 653 (w), 543 (s), 508 (s).

#### 3.4.2. [Zn_2_L_2_]·nC_7_H_8_ (1)

To a solid H_2_L (0.150 g, 0.367 mmol), anhydrous ZnCl_2_ (0.500 g, 0.367 mmol) and KHMDS (0.146 g, 0.734 mmol), THF (10 mL) was added. The mixture was stirred overnight under heating at 60 °C. The THF was removed by evaporation. The substance was dissolved in toluene and the KCl precipitate was removed by centrifugation. Evaporation of the solvent to a minimum volume and periodic heating and cooling of mixture led to the formation of suitable for XRD analysis dark red crystals. The solid was centrifuged and washed with toluene (3 × 2 mL) and dried under vacuum. Yield 0.0715 g (40%). Calc. for C_36_H_22_N_12_P_2_S_4_Zn_2_·0.1C_7_H_8_ (%): C 46.3; H 2.4; N 17.6; S 13.5; found: C 46.5; H 2.6; N 17.5; S 13.6. ^1^H NMR (C_6_D_6_, δ, ppm): 6.36 (d, 1H), 6.53 (t, 1H), 6.65 (d, 1H), 6.75 (t, 2H), 6.83 (dd, 1H), 6.95 (d, 1H), 7.03 (dd, 1H), 7.21 (qu, 1H), 7.76 (t, 2H). ^31^P{H} NMR (C_6_D_6_, δ, ppm): 74.1 (s). IR (cm^−1^): 3044 (m), 2850 (w), 1603 (w), 1568 (m), 1528 (s), 1473 (s), 1437 (w), 1374 (s), 1286 (s), 1170 (w), 1099 (s), 1041 (w), 905 (s), 885 (s), 851 (m), 806 (m), 739 (s), 699 (w), 671 (w), 543 (w), 508 (m).

#### 3.4.3. [Zn_2_(H_2_L)_2_Cl_4_]·nC_7_H_8_ (2a·nC_7_H_8_ and 2b·nC_7_H_8_)

To a solid H_2_L (0.0749 g, 0.183 mmol) and anhydrous ZnCl_2_ (0.0250 g, 0.183 mmol), THF (10 mL) was added. The red solution was stirred overnight and the solvent was removed off by evaporation. The red fine-crystalline precipitate was obtained by slow extraction with toluene (5 mL) in a two-sector ampoule. The obtained crystals were suitable for XRD analysis. The solution was decanted, the crystals were washed with toluene (3 × 2 mL) and dried under vacuum. Yield 0.1067 (95 %). Calc. for C_36_H_26_Cl_4_N_12_P_2_S_4_Zn_2_·1.1C_7_H_8_ (%): C 44.1; H 2.9; N 14.1; found: C 43.7; H 3.1; N 13.5. IR (cm^−1^): 3353 (m), 3290 (m), 3055(w), 2918 (m), 2849 (m), 1602 (w), 1549 (s), 1487 (s), 1364 (m), 1289 (m), 1086 (s), 1077 (w), 1047 (w), 916 (w), 871 (w), 825 (w), 744 (m), 702 (w).

#### 3.4.4. [Cu(H_2_L)Cl]_n_ nTHF (3)

A solution of H_2_L (0.0202 g, 0.0495 mmol) in THF (5 mL) was added to a vial with powder of CuCl (0.0049 g, 0.049 mmol). The mixture was left without stirring in the closed vial. A week later, yellow plate crystals formed. The obtained crystals were suitable for XRD analysis. The solution was decanted, the crystals were washed with THF (3 × 2 mL) and dried under vacuum. Yield 0.0176 g (60%). Calc. for C_18_H_13_ClCuN_6_PS_2_·C_4_H_8_O (%): C 45.6; H 3.6, N 14.5; found: C 45.4; H 3.7; N 14.4. IR (cm^−1^): 3309 (w), 3071 (w), 2973 (w), 2856 (w), 1609 (m), 1539 (s), 1490 (s), 1435 (m), 1379 (s), 1299 (m), 1161 (w), 1082 (m), 1049 (w), 972 (w), 907 (m), 857 (m), 828 (w), 804 (w), 747 (s), 694 (w), 667 (w), 545 (w), 520 (w).

## 4. Conclusions

To conclude, we present the synthesis of the novel phosph(III)azane (H_2_L) based on 2,1,3-benzothiadiazole. The presence of accessible lone pairs on both P and N atoms makes this phosph(III)azane a promising multidentate ligand for transition metal complexes. The coordination ability of this derivative and its deprotonated species (L^2–^) is demonstrated on the example of Zn and Cu(I) complexes **1**–**3** that were isolated as the solvates with toluene or THF. According to single crystal XRD analysis, the complexes of the formula [Zn_2_(H_2_L)_2_Cl_2_] can exist in two isomeric species, *viz*. **2a** and **2b**, that differ by the arrangement of one of H_2_L ligand. The quantum chemical calculations reveal these complexes are thermodynamically equally favorable. Inside the molecules **2a** and **2b**, tetragonal prismatic cavities are observed that large enough for the inclusion of solvate toluene molecule. The toluene arranged differently inside the complex molecules defining their geometry. With the removal of toluene, the geometry of **2b** changes substantially, while that of **2a** changes slightly. The side product of the synthesis of H_2_L, *viz*. four-membered cyclophosphazane **4**, has been characterized by single crystal XRD analysis. The photophysical properties of H_2_L **1**–**3** were studied. H_2_L and the mixture of the phases **2**·nC_7_H_8_ reveal fluorescence in the visible range; the single-peak band and the single-exponential decay kinetics for **2**·nC_7_H_8_ indicates the species **2a** and **2b** possess similar photophysical properties. The difference in the absorption spectra of H_2_L, **1** and **2b** is discussed using TD-DFT calculations. These compounds reveal ligand-centered low-energy transitions; their charge transfer nature in H_2_L and **2b** is similar, while that in **1** differs significantly from them due to anionic nature of L^2–^ and different coordination type of the ligand.

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
