# Peer review of "Structural and Photophysical Properties of 2,1,3-Benzothiadiazole-Based Phosph(III)azane and Its Complexes"

_molecules, 2020, doi:10.3390/molecules25102428_

Round 1

Reviewer 1 Report

The synthesis and the characterization of phosphorus-containing benzothiadiazoles and of the corresponding metal complexes is an interesting field of the coordination chemistry, aiming to develop new functional materials. For these reasons, this manuscript is worth of publishing but after some minor revisions due to clarify the points above. In particular, in order to confirm the hypothesis regarding the differences observed in the absorption spectrum of complex 1, could the authors perform TD-DTF calculations on this complex as well as on the complexes 2 and 3?

Author Response

In particular, in order to confirm the hypothesis regarding the differences observed in the absorption spectrum of complex 1, could the authors perform TD-DTF calculations on this complex as well as on the complexes 2 and 3?

We performed TD-DFT calculations for compounds 1 and 2b. Unfortunately, we were unable to make TD-DFT calculations for compound 3 due to its complicated periodic structure; such calculations require much longer time. We assume that 2a shows similar to 2b electronic structure and thus shows similar photophysics; for this reason, TD-DFT calculations for 2a were also not performed. However, we believe that a comparison of the calculated data for 1 and 2b is sufficient in the article context. Based on the calculations, we updated the “Photophysical Properties” as well as “Abstract” and “Conclusion” sections (changes are marked green). We also included Tables S2, S3 and Figure S7 in the new version of Supporting Information.

Reviewer 2 Report

The manuscript presents a very complete and consistent study on the synthesis and optical properties of N,N'-bis(2,1,3-benzothiadiazol-4-yl)-1-phenylphosphanediamine (H2L) and its zinc (II) and copper (I) coordination compounds. The authors show that H2L ligand and its deprotonated species exhibit different coordination modes. They are also reporting the existence of two isomeric forms of the complexes of Zn (II) with H2L. The discussion of the results is systematic and well organized. The conclusions are well supported by the discussion. The materials and method section provide a description of the experiments in good enough detail to ensure reproducibility of results. The overall readability of the manuscript is good. I believe the manuscript is suitable for publication once the authors address the following issues:

  • Check the consistency of the 2a and 2b nomenclature of the two isomeric forms throughout the manuscript. It appears that the nomenclature in figures 5 and 6 differs from that in Scheme 2

Author Response

Check the consistency of the 2a and 2b nomenclature of the two isomeric forms throughout the manuscript. It appears that the nomenclature in figures 5 and 6 differs from that in Scheme 2

We revised Scheme 2: phosphorus - phenyl bonds are now identified by wedges. This may help to understand the nomenclature of 2a and 2b. We also included the solvate molecules into the formula of the compounds in Scheme 2. Throughout the manuscript, we use the formulae “2a·3C7H8” and “2b·2.5C7H8” derived from XRD when discuss the crystal structures; the formulae “2a·nC7H8” and “2b·nC7H8” are for the dried phases (individual compounds) which lost almost all toluene molecules. We use “2a” and “2b” to denote the single molecules (complexes). Similar abbreviation is used for other compounds/molecules.